# Impact of κ-Carrageenan on the Cold-Set Pea Protein Isolate Emulsion-Filled Gels: Mechanical Property, Microstructure, and In Vitro Digestive Behavior

**DOI:** 10.3390/foods13030483

**Published:** 2024-02-02

**Authors:** Xiaojiao Li, Xing Chen, Hao Cheng

**Affiliations:** 1State Key Laboratory of Food Science and Resources, Jiangnan University, Wuxi 214122, China; 6210112042@stu.jiangnan.edu.cn; 2School of Food Science and Technology, Jiangnan University, Wuxi 214122, China; 3State Key Laboratory of Marine Food Processing & Safety Control, Qingdao 266400, China

**Keywords:** emulsion-filled gel, pea protein isolate, κ-carrageenan, mechanical property, microstructure, simulated in vitro digestion

## Abstract

More understanding of the relationship among the microstructure, mechanical property, and digestive behavior is essential for the application of emulsion gels in the food industry. In this study, heat-denatured pea protein isolate particles and κ-carrageenan were used to fabricate cold-set emulsion gels induced by CaCl_2_, and the effect of κ-carrageenan concentration on the gel formation mechanism, microstructure, texture, and digestive properties was investigated. Microstructure analysis obtained by confocal microscopy and scanning electron microscopy revealed that pea protein/κ-carrageenan coupled gel networks formed at the polysaccharide concentration ranged from 0.25% to 0.75%, while the higher κ-carrageenan concentration resulted in the formation of continuous and homogenous κ-carrageenan gel networks comprised of protein enriched microdomains. The hydrophobic interactions and hydrogen bonds played an important role in maintaining the gel structure. The water holding capacity and gel hardness of pea protein emulsion gels increased by 37% and 75 fold, respectively, through increasing κ-carrageenan concentration up to 1.5%. Moreover, in vitro digestion experiments based on the INFOGEST guidelines suggested that the presence of 0.25% κ-carrageenan could promote the digestion of lipids, but the increased κ-carrageenan concentration could delay the lipid and protein hydrolysis under gastrointestinal conditions. These results may provide theoretical guidance for the development of innovative pea protein isolate-based emulsion gel formulations with diverse textures and digestive properties.

## 1. Introduction

Emulsion-filled gels are a class of semi-solid materials that limit the emulsified oil droplets within the gel network, possessing the advantages of both emulsions and hydrogels [1,2]. Emulsion-filled gels are commonly presented in many traditional food systems, such as cheese, yogurt, tofu, mayonnaise, and some other dairy desserts [3,4,5]. They also can be used as a solid fat substitute and edible inks for 3D printing techniques to create novel and healthy products [6,7]. In the food industry, the advantages of utilizing emulsion-filled gels are that they can improve the kinetic stability, sensory, and texture of existing food matrix and modulate the chemical stability, release profile, and bioavailability of functional ingredients during the storage and digestion [8,9]. The mechanical properties and digestive behavior are key factors affecting the functional performance of emulsion-filled gels in the food industry, which can be tailored by selecting different structural elements and fabrication methods.

Pea protein isolate (PPI), consisting mainly of 65~80% globulins and 10~20% albumins, is a new alternative protein in the food industry due to its hypoallergenic, high nutritional values, low cost, and multiple functionalities [10,11]. Heat treatment is a traditional method for the fabrication of pea protein gels, resulting from protein unfolding, exposure of active side chains, and protein/protein interaction during the thermal process [12,13]. However, many studies pointed out the weaker gelling capacity of pea proteins and lower texture features of pea protein heat-set gels compared to soy proteins and whey proteins [14,15,16]. Furthermore, the heat-induced pea protein gels would require high protein concentration (above 10 wt%) and high-concentration salts (1.0~2.0 wt% NaCl) [13,17].

Recently, the pH-shifting pre-treatment was able to improve the mechanical properties of pea protein gels because of partial protein unfolding and crosslinked protein aggregates formation before the heat-treatment gelation [18]. Meanwhile, the cold-set gelation could be used as an alternative route for the fabrication of pea protein emulsion gels at a relatively low pea protein concentration. The heat denaturation of pea protein is obtained at a pH far from its isoelectric point and in the absence of salt, and then the gelation is achieved by adding gelling agents including ions and acids [19,20,21]. Besides the advantage of promoting the functionalities of protein gels, cold-set gelation could control soluble protein aggregation and encapsulate heat-sensitive ingredients compared to heat-induced gelation [22,23].

Another strategy for the modification of pea protein gel texture and functionality is to use the protein–polysaccharide interactions. Previous studies have reported that pea proteins were complexed with many polysaccharides to achieve improved emulsifying and gelling properties [21,24,25]. κ-Carrageenan (KC) is a linear polysaccharide with one sulfate group in each repeating sugar unit, consisting of α-(1-3)-D-galactopyranose and β-(1-4)-3,6-anhydro-D-galactopyranose [26]. In the system where pea proteins and KC coexist, phase separation or association occurs when there is strong repulsion or strong attraction between them, which can modulate microstructures and mechanical properties of heat-induced pea protein gels [27]. For example, the addition of 0.5% KC into 7.5% heat-denatured pea protein aggregates could prevent the large aggregate precipitation and make more ordered protein aggregate association during gel formation, which could significantly improve the mechanical strength of heat-induced gels with a value of around 14 kPa [28]. However, the research effort on the effect of KC on the pea protein-based emulsion gels is still limited.

Several factors affect the structure of emulsion-filled gels and their mechanical and digestive properties, such as the properties of matrix materials, gelation methods, and the interaction of droplets with the gel matrix [29]. Understanding the relationship among microstructure, mechanical properties, and digestive behavior of emulsion-filled gel systems facilitates the development of novel food formulations with multiple physiological properties including satiety enhancement, bioavailability improvement, and consumer health and nutrition promotion. In recent studies, pea proteins have been used as building blocks for the fabrication of emulsion-filled gels using a cold-set gelation method to clarify the gel formation mechanism and property [30,31]. However, there has not been a systematic investigation on the structure–property relationships of pea protein-based emulsion gels affected by the addition of KC. In this paper, the emulsion-filled gel was fabricated with pea proteins using a cold-set gelation method. The effect of KC on changes in the microstructure, mechanical properties, and digestive behavior of emulsion gels was clarified. We hope to provide a theoretical basis for the design of emulsion gels for application in the food industry by constructing a relationship among microstructure, mechanical properties, and digestive behavior.

## 2. Materials and Methods

### 2.1. Materials

Commercial pea protein flour (protein content ~84%) was purchased from Xi’an Binghe Biotechnology Co., Ltd. (Xi’an, China). Bile salts, Nile Red, Nile Blue, κ-carrageenan (product number 22048), β-mercaptoethanol, fluorescein-5-isothiocyanate (FITC), sodium dodecyl sulfate (SDS), porcine pancreatin (4 × USP specifications), and porcine pepsin (≥500 U/mg) were purchased from Sigma-Aldrich Co., Ltd. (St. Louis, MO, USA). Sunflower oil (Brand Duoli) was obtained from a local retailer (Wuxi, China). Other agents were of analytical grade and obtained from SinoPharm CNCM Ltd. (Shanghai, China). Deionized water produced from a Milli-Q water purification system (Millipore, Billerica, MA, USA) was used for solutions and emulsions preparation.

### 2.2. Pea Protein Isolate (PPI) Extraction

Commercial pea protein flour was further purified using the alkaline extraction followed by the isoelectric precipitation method [32]. Commercial pea protein flour was dispersed in deionized water in a flour-to-water ratio of 1:40 (*w*/*v*). Then, the pH of the mixture was adjusted to pH 9.0 and stirred for 1 h at room temperature followed by centrifugation at 5000× *g* for 15 min at 10 °C. The supernatant was adjusted to pH 4.5 and precipitation was collected by centrifugation at 8000× *g* for 15 min at 10 °C. The obtained precipitate was dispersed in deionized water at pH 7.0. The solution was freeze dried using a Freeze Dryer (Labconco, Kansas City, MO, USA) for 72 h to obtain the PPI with a protein content of 94.6 ± 1.1 g/100 g (%N × 6.25).

### 2.3. Emulsion Preparation

PPI aggregates were prepared using a combination of pH-shifting and heating treatment methods [18,28]. PPI solution (5%, *w*/*w*) was titrated to pH 12.0 by 6 M NaOH for 1 h and then titrated back to 7.0 by 6 M HCl. The PPI solution was further allowed to heat at 90 °C for 30 min to induce PPI aggregation. κ-Carrageenan (KC) stock solution (3%, *w*/*w*) was prepared by dissolving the powder in deionized water at 55 °C. Emulsions were prepared by blending the PPI aggregate dispersion and sunflower oil (10%, *w*/*w*) at 10,000 rpm for 2 min using a high-speed blender (Ultra Turrax T25, IKA, Staufen, Germany), followed by high-pressure homogenization (AH-2010, ATS Engineering Ltd., Brampton, ON, Canada) at 50 MPa for three passes. The prepared emulsion was mixed with different volumes of KC solution and deionized water and stirred at 55 °C for 30 min. Eventually, emulsions contained 2.5% (*w*/*w*) PPI aggregates, 10% (*w*/*w*) sunflower oil, and 0~1.5% (*w*/*w*) KC.

### 2.4. Emulsion Size Distribution and ζ-Potential Measurement

Size distribution and ζ-potential of emulsion samples determined by a dynamic light scattering instrument (Brookhaven Instruments Ltd., New York, NY, USA) at 25 °C before the appropriate dilution. The size distribution was expressed as the intensity distribution obtained at a scattering angle of 90°.

### 2.5. Cold-Set Emulsion-Filled Gel Preparation

CaCl_2_-induced emulsion-filled gels were fabricated followed by a previously reported method [33]. The emulsions in the absence and presence of KC were rapidly mixed with 1 M CaCl_2_ solution and stored overnight at 4 °C for gel formation. The final concentration of CaCl_2_ was 40 mM in the gels.

### 2.6. Emulsion-Filled Gel Properties Characterization

#### 2.6.1. Water Holding Capacity (WHC)

The WHC of emulsion gels was determined by a previously reported method [29]. About 3 g gel samples were placed into 50 mL tubes and then centrifuged at 8000× *g* for 20 min. Any excess water was removed using filter paper. The WHC was calculated using the following Equation (1):(1)$$\mathrm{WHC}\;(\%)\;=\left(1-\frac{W_{t}-W_{1}}{W_{t}}\right)\times 100\%$$
where *W*_t_ represents the mass of the emulsion gel before centrifugation (g), and *W*_1_ represents the mass of the emulsion gel after centrifugation to remove water (g).

#### 2.6.2. Gel Texture Profile Analysis

The analysis texture profile of emulsion gels was investigated using an instrumental texture analyzer (TA. XTPlus, Stable Micro Systems, Surrey, UK) equipped with a P-36R cylindrical test probe [34]. The cylindrical emulsion gel samples with a diameter of 2 cm and height of 1.5 cm were placed on the carrier table for texture profile analysis through a 2-cycle sequence with a strain of 50% at a trigger force of 3 g and a test speed of 1.0 mm/s. The textural parameters were provided by the accompanying software.

### 2.7. Emulsion Gel Microstructure Characterization

#### 2.7.1. Confocal Laser Scanning Microscopy (CLSM)

The microstructure of emulsion gels and gastrointestinal digesta was visualized by a CLSM (LSM710, Leica, Heidelberg, Germany). Nile Red (2 mg/mL) and Nile Blue (2 mg/mL) were used to physically label the oil phase and protein of emulsion gels, respectively. KC was covalently labeled by fluorescein-5-isothiocyanate (FITC) [35]. Excitation wavelengths of 488 nm, 514 nm, and 633 nm were used for the FITC channel, Nile Red channel, and Nile Blue channel, respectively.

#### 2.7.2. Scanning Electron Microscopy (SEM)

The emulsion gel sample was immobilized in 2.5% (*v*/*w*) glutaraldehyde at 4 °C for 24 h. After this, the sample was washed with 0.1 M phosphate buffer (pH 7.0) for 3 times. Then, the gel sample was freeze dried at −86 °C for 72 h. The dried gel sample was cut into 5 mm × 5 mm × 2 mm (long × width × height) pieces with a blade, and its microstructure was observed by SEM (SU8100, Hitachi, Tokyo, Japan) at 60×, 300×, and 1000×, respectively, after sputtering gold.

### 2.8. Fourier Infrared Spectroscopy (FT-IR)

Pea protein emulsion-filled gels at various concentrations of KC were ground into powders after freeze drying. The sample (1.0 mg) was pressed into lamellas by mixing with 100.0 mg of KBr. FT-IR spectra (resolution = 4 cm^−1^, 64 scans) were recorded on a Nicolet IS10 spectrometer (Thermo Electron Co., Waltham, MA, USA) at 25 °C with pure KBr as the background. The protein amide I region (1700~1600 cm^−1^) was processed by Fourier self-deconvolutions with OMNIC 9.2 software with a bandwidth of 24 cm^−1^ and an enhancement factor of 2.5 [36].

### 2.9. Measurement of Protein Solubility

The solubility of emulsion gels was measured in different solvents according to the previously reported protocol with slight modifications [37]. The gel sample (1.0 g) was dissolved in 5 mL of different solvents under continuous stirring for 3 h at 25 °C or 60 °C. The solvents were distilled water, Tris-glycine buffer containing 4 mM Na_2_•ethylene diamine tetraacetic acid (EDTA) (0.086 M Tris, 0.09 M glycine, pH 8.0), 2% (*w*/*v*) sodium dodecyl sulfate (SDS), 8 M urea, and 1% (*v*/*v*) β-mercaptoethanol. The protein concentration was determined using the Kjeldahl method.

### 2.10. In Vitro Digestion of Emulsion Gels

#### 2.10.1. Simulated Gastrointestinal Tract Digestion

Emulsion gels were digested by subjecting them to sequential incubation in simulated saliva fluid (SSF, pH 7.0), simulated gastric fluid (SGF, pH 3.0), and then simulated intestinal fluid (SIF, pH 7.0) using the INFOGEST 2.0 protocol with a slight modification [38]. Briefly, 10 g of each emulsion gel was shredded and mixed with 10 mL of SSF at pH 7.0 for 3 min. Then, the mixture was incubated with 20 mL SGF at pH 3.0 for 2 h. After that, the mixture was further mixed with 40 mL SIF for 4 h. The pH of the simulated intestinal digestion system was maintained at 7.0 using 0.1 M NaOH throughout the digestion period. The release of free fatty acids (FFA) can be calculated by the Equation (2) [9,39]. All digestive procedures were conducted at 37 °C. The formulation of digestive fluids was described as follows: SSF consisted of 15.1 mM KCl, 13.6 mM NaCl, 1.5 mM CaCl_2_(H_2_O)_2_ at pH 7.0; SGF consisted of 6.9 mM KCl, 72.2 mM NaCl, 0.15 mM CaCl_2_(H_2_O)_2_, and 2000 U/mL pepsin at pH 3.0; SIF consisted of 6.8 mM KCl, 123.4 mM NaCl, 0.6 mM CaCl_2_(H_2_O)_2_, 10 mM bile salts, and 100 U/mL pancreatin.
(2)$$\mathrm{FFA}(\%)\;=\frac{V_{\mathrm{NaOH}}\times C_{\mathrm{NaOH}}\times M_{\mathrm{Lipid}}}{{2\times W}_{\mathrm{Lipid}}}\times 100\%$$
where *V*_NaOH_ is the volume of NaOH solution consumed during the titration (mL), *C*_NaOH_ is the molar concentration of NaOH solution used for the titration (0.1 M), *M*_Lipid_ is the average molecular weight of sunflower oil (880 g/mol), and *W*_Lipid_ is the weight of lipid in the digested samples (g). Each triacylglycerol can be decomposed into two free fatty acids by lipase.

#### 2.10.2. Measurement of Oil Droplet Size

The digesta at different periods were obtained and sieved before enzyme inactivation. SDS (2 wt%) and dithiothreitol (10 mM) were added to digestive fluids for the liberation and stabilization of oil droplets [40]. The particle size of the oil droplets was measured using the particle size analyzer (S3500, Microtrac Co., Ltd., Pennsylvania, UK). The refractive index for emulsified oil droplets was 1.47. The size of oil droplets was expressed as the average volume-weighed diameter (M_V_).

#### 2.10.3. Degree of Protein Hydrolysis

The degree of hydrolysis (DH) was measured according to the ninhydrin colorimetric method with some modifications [41]. The digestive fluids of the stomach and intestine with different digestion times were heated in boiling water for 1 min to inactivate the digestive enzymes and then centrifuged at 14,500× *g* for 15 min. The supernatant was mixed with distilled water and ninhydrin color reagent (3.0 M anhydrous Na_2_HPO_4_, 0.5 g ninhydrin, and 0.3 g fructose in 100 mL distilled water). The mixtures were heated in a boiling water bath for 15 min. After diluting the mixtures with 40% ethanol, the absorbance of the mixtures under 575 nm was determined. The concentration of amino acid was calculated by using the standard curve of glycine and Equation (3).
(3)$$\mathrm{DH}\left(\%\right)=\frac{h}{h_{\mathrm{tot}}}\;\times \;100\%\;$$
where *h* is the number of broken peptide bonds in 1.0 g protein, mmol/g, *h*_tot_ is the number of peptide bonds in 1.0 g pea protein (7.55 mmol/g), mmol/g.

### 2.11. Statistical Analysis

Each sample was measured independently in all tests and all data were expressed as mean ± standard deviation (SD). The significance of the differences was determined by a one-way analysis of variance (ANOVA) and Dunnett’s test, which was analyzed by the software SPSS 20.0. The value of *p* < 0.05 was considered as a significant difference.

## 3. Results and Discussion

### 3.1. Size Distribution and ζ-Potential of Emulsions

The PPI aggregate stabilized O/W emulsion had a bimodal size distribution with peaks around 82 nm and 357 nm (Figure 1A). The bimodal size distribution was commonly observed in protein particle-stabilized O/W emulsions, such as whey protein microgel particle-stabilized O/W emulsions [42], and zein nanoparticle-stabilized peppermint oil emulsions [43]. The smaller peak was attributed to the PPI aggregates (Appendix A), and the larger peak was the emulsified oil droplets. The addition of KC led to a slight increase in the size of both PPI aggregates and emulsified oil droplets, as the polysaccharide concentration increased up to 0.75%. When the KC concentration further increased up to 1.5%, the size of emulsified oil droplets sharply increased to around 2 μm. In this situation, the presence of excessive KC molecules in the continuous phase caused the massive aggregation of emulsified oil droplets due to depletion flocculation [44,45]. At this point, the presence of massive non-adsorbed KC molecules in the continuous phase would induce an osmotic gap between the depletion zone around the emulsified oil droplets and the continuous phase, because water molecules in the depletion zone tend to diffuse into the continuous phase to reduce the osmotic gap, thereby increasing the aggregation of emulsified oil droplets that occurs [44].

The ζ-potential of the PPI aggregate stabilized O/W emulsion was −39 mV at pH 7.0 (Figure 1B). The net negative charges of emulsified oil droplets stabilized by pea protein aggregates were due to their isoelectric point between 4.0 and 5.0 [46]. All emulsions had a ζ-potential value of around −40 mV as the KC concentration varied from 0.25% to 1.5%, indicating that the addition of KC possessed negligible effect on the surface charges of emulsified oil droplets. It is therefore suggested that KC molecules were not adsorbed onto the surface of emulsified oil droplets at pH 7.0, attributing to the electrostatic repulsion between the negatively charged anionic KC and emulsified oil droplets [47,48]. Thus, most of the KC molecules remained free in the continuous phase of the emulsions.

### 3.2. Emulsion Gel Formation and Microstructure

#### 3.2.1. Confocal Laser Scanning Microscopy (CLSM)

All formulations in this study could form the self-supported emulsion-filled gels, and the co-location of protein/oil and protein/polysaccharide in emulsion-filled gels were visualized by CLSM in Figure 2A and Figure 2B, respectively. As shown in Figure 2A, Ca^2+^-induced pea protein emulsion-filled gels possess a heterogeneous, porous, and loose gel network. The oil fraction was co-localized with pea protein, indicating that emulsified oil droplets were emended within the protein gel matrix. The addition of 0.25% KC decreased the spatial heterogeneity of the protein network, indicating that the presence of KC could prevent the extensive aggregation of PPI particles and emulsified oil droplets. As the KC concentration was 0.75%, the more dense and aggregated protein gel network formed. However, when the KC concentration was above 1.0%, the protein network became less homogenous, and increased microdomains enriched with protein and oil droplets were observed.

As shown in Figure 2B, KC was co-localized with pea protein in the emulsion gel as the KC concentration was below 0.75%. As the KC concentration gradually increased, the pea protein/KC gel network became more compact with thicker strands. These results suggested that the emulsion gels possess protein/polysaccharide coupled gel networks. As the KC concentration was above 1.0%, the continuous and homogenous KC networks dominated the emulsion gel structure, and incomplete pea protein aggregates were distributed within the polysaccharide gel network.

Ca^2+^ would be expected to induce protein aggregates crosslinking, ion complexation between pea protein and KC, and KC gelation [22,49,50]. In the absence of KC, Ca^2+^ could crosslink PPI aggregates and emulsified oil droplets quickly, resulting in the formation of microstructure presented in the form of random aggregation, clumps, or clusters (Figure 2A). This gel network was typically observed in protein-based (e.g., whey proteins and soy proteins) emulsion-filled gels prepared by the Ca^2+^-mediated cold gelation method [51,52]. Once high negatively charged KC was added at a concentration of 0.25%, Ca^2+^ would preferentially interact with the negative charges on the surface of KC molecules [53]. Ca^2+^ favored the coil–helix transition of KC molecules and ion complexation between pea protein and KC [49], resulting in the formation of pea protein/KC coupled gel networks (Figure 2B). The coupled gel network formation could prevent pea protein particles and emulsified oil droplets from extensive aggregation, which is consistent with previous obversion in Ca^2+^-induced soy protein isolate/konjac gum emulsion gels [54]. With the further increase in the KC concentration to 0.75%, these interactions were strengthened and formed more compact and dense coupled gel networks. It should be noted that more fibrillar aggregates could be observed at high KC concentration, due to the assembly of KC molecules to supramolecular strands induced by Ca^2+^ [55]. As the KC concentration was above 1.0%, the microphase separation occurred and protein-enriched microdomains formed. Meanwhile, the high level of KC is sufficient to form a gel network in the aqueous phase mediated by Ca^2+^, thereby forming a KC uniform gel network entrapped with pea protein and emulsified oil droplets [50].

#### 3.2.2. Scanning Electron Microscopy (SEM)

The microstructure of emulsion gels was further observed by SEM (Figure 3). Pea protein emulsion gels possessed a lamellar and honeycomb combined structure with small and uniform pores and smooth surfaces. The addition of KC could significantly affect the structural characteristics of pea protein emulsion gels. As the concentration of KC was below 0.5%, the emulsion gels possessed an interconnected honeycomb structure with disordered and heterogeneous pores, which is consistent with the observation in alginate/casein emulsion gels [56]. Meanwhile, the surface of the gels network became rough and possessed a unique ridge-like structure of KC gels, on which the oil droplets emulsified by pea protein aggregates grew uniformly [57]. These results suggested that KC in the aqueous phase formed a gel network, which is consistent with CLSM results (Figure 2). The diversity in the pore size of emulsion gels was mainly because the low gel strength could not maintain the pore size. At the same time, the presence of smaller pores in the gel networks indicated that the pea protein gel networks and KC gel networks were not intertwined with each other completely [58]. When the concentration of KC reached 0.75%, there was a structure of small pores around large pores in the gel network, indicating that the two networks of proteins and polysaccharides were connected and intertwined. Meanwhile, smoother surfaces and thicker wall layers of emulsion gel networks were observed as the further increased KC concentration.

### 3.3. Pea Protein Conformational Change

The protein conformation of pea protein during the heat treatment and gelation was investigated by FTIR spectrum. Figure 4 shows the deconvoluted amide I band (1700~1600 cm^−1^), which indicates the changes in the secondary structure of pea proteins. For the native pea protein, the following peaks could be assigned as follows: near 1608 cm^−1^ (amino acid residue vibrations), near 1621~1635 cm^−1^ and 1693 cm^−1^ (β-sheets), near 1648 cm^−1^ and 1660 cm^−1^ (α-helices), near 1669 cm^−1^ and 1680 cm^−1^ (β-turns), and near 1639 cm^−1^ (unordered structures) [28]. After the pH-shifting and heating treatment, the pea protein went through the process of structural expansion to refolding. After refolding, the peaks of pea protein around 1648 cm^−1^ and 1660 cm^−1^ increased, indicating that the α-helix structure increased, and the thermal denaturation process formed a denser fold. However, compared with the native pea protein, the position of the α-helix absorption peak is shifted, which may be due to the destruction of the original hydrogen bond and the promotion of intermolecular interaction through thermally reversible hydrogen bonds [17]. At the same time, a new absorption peak appeared at 1667 cm^−1^, indicating that more β-turns were formed during the folding process of pea protein.

In the process of formation induced by the addition of calcium ions, the absorption peak at 1647 cm^−1^ increased obviously, and more α-helical structures were formed. At the same time, a new absorption peak appeared near 1625 cm^−1^, which indicated that the pea protein formed more β-sheet structure. With the increase in the concentration of κ-carrageenan, the intensity of these two absorption peaks increased, indicating that the existence of KC increased the content of α-helix and β-sheet, which was consistent with the reports in the literature [59]. This is mainly caused by the conversion of β-turn and random coil [60].

### 3.4. Emulsion Gel Dissociation Test

Electrostatic interactions, hydrophobic interactions, hydrogen bonds, and disulfide bonds are commonly involved in the protein/protein, polysaccharide/polysaccharide, and protein/polysaccharide interactions during the emulsion gel formation, which could be destroyed by EDTA, SDS, urea, and β-mercaptoethanol, respectively [61]. Figure 5A,B shows the visual appearance of emulsion gels and protein solubility in different solvents at 25 °C and 60 °C, respectively. Pea protein emulsion gels melted and lost their shape in all types of solvents within 3 h at 25 °C (Figure 5A), indicating that they possessed poor stability and lacked self-supporting ability in the aqueous phase. With increasing KC concentration, pea protein/KC emulsion gels retained their shapes except in SDS and urea solutions, indicating that the hydrophobic interactions and hydrogen bonds played important roles in the gel network formation. Noticeably, more extensive gel melt was observed at 60 °C, especially in pea protein/KC emulsion gels with 1.5% KC (Figure 5B). This was attributed to the partial destruction of hydrogen bonds and the gel–sol transition of KC induced by the increased temperature [62]. These results also confirmed that KC gel networks dominated the emulsion gel structure at the high KC concentration (Figure 2).

As shown in Figure 5C, the protein solubility of pea protein emulsion-filled gels is independent of the solvent type, possibly due to the weak gel strength and poor gel stability. When the KC concentration was less than 0.75%, the protein solubility decreased in the individual solvent as the polysaccharide concentration increased. It has been reported that all these non-covalent and covalent interactions may be involved in the protein–protein and protein–polysaccharide interplays of pea protein/polysaccharide emulsion gel formation [30]. Therefore, the sole solvent could not extract pea protein from the emulsion gels efficiently. However, the protein solubility in SDS and urea significantly increased as the KC concentration was greater than 0.75%. The increased temperature caused the increased pea protein solubility in different solvents, especially in SDS and urea (Figure 5D). These results suggested that the main forces to maintain the pea protein emulsion gels with high KC concentration were hydrogen bonds and hydrophobic interactions.

### 3.5. Water Holding Capacity (WHC)

In Figure 6, the WHC of pea protein emulsion-filled gels was around 60%. The pea protein emulsion-filled gels showed higher water holding capacity than Ca^2+^-induced emulsion gels formulated with 5% denatured whey protein isolate and 10% sunflower oil with a value of around 50% [33]. With increasing the concentration of KC to 0.75%, the WHC of emulsion-filled gels gradually increased to around 96%. The further increase in the KC concentration exhibited no significant impact on the WHC of emulsion gels, attributed to the fact that the high KC concentration did not damage the gel structure (Figure 3). The increased KC concentration favored the formation of pea protein/KC coupled or continuous KC dense gel networks with reduced pore size (Figure 2), thereby improving stronger capillary forces for the retention of more water molecules [63,64]. In addition, the large numbers of hydroxyl groups on the KC can interact with water molecules by hydrogen bonding and reduce water loss [34,65].

### 3.6. Textural Properties

Textural properties of pea protein emulsion gels at various concentrations of KC are shown in Table 1. The pea protein emulsion-filled gels induced by Ca^2+^ possessed a soft texture with a hardness value of 21 g, which is similar to the previous report on Ca^2+^-induced whey protein emulsion-filled gels [33,63]. The addition of KC could increase the hardness of pea protein emulsion gel by around 2 fold and 10 fold at the KC concentration of 0.25% and 0.5%, respectively. The hardness of emulsion gels reached 1560 g as the KC concentration increased up to 1.5%. The chewiness of emulsion gels had a similar pattern to the hardness since more energy is required for chewing associated with the increased hardness of emulsion gels [66]. The addition of KC could improve the gel springiness but did not show a considerable impact on the cohesiveness of pea protein emulsion gels (Table 1). A similar phenomenon was previously reported in Ca^2+^-induced egg yolk/sodium alginate emulsion gels [64] and whey protein isolate/basil seed gum emulsion gels [67].

The hardness and chewiness of emulsion gels are dependent on the particulate content of solids and structures of the gel matrix and emulsified oil droplets [68,69]. The addition of KC favored the dense and thick gel network formation and improved the steric exclusion effect. The high KC concentration induced microphase separation due to the thermodynamic incompatibility, increasing the regional effective protein concentration and protein/protein interactions during the gel formation [70]. Additionally, the KC self-gel may synergistically interact with pea protein for the enhancement of the gel harness [71]. For the springiness and cohesiveness, the emulsion gel exhibited a profile of increasing first and then decreasing as the KC concentration increased. The main reason is that excessive addition of KC will lead to the formation of depleted flocculation of emulsion oil droplets, influencing the uniformity of the gel, and thus negatively affecting the cohesiveness and resilience of the gels [64].

### 3.7. Gastrointestinal Fate of Emulsion Gels

#### 3.7.1. Microstructure of Digesta

CLSM images of digesta at the gastric and intestinal stages are shown in Figure 7. After 2 h of gastric digestion, pea protein emulsion gels without KC were mostly broken down, which consisted of gel particles of various sizes. The rapid breakdown of bulk emulsion gels in the gastric stage was previously observed in whey protein emulsion gels, attributed to the mechanical shred, enzymatic hydrolysis of the protein gel matrix, and soft gel texture [72,73]. The addition of KC resulted in the larger gel particles remaining after the gastric digestion, which was more pronounced at the high KC concentration. During in vitro gastric digestion, oil droplets were relatively stable and almost retained in the protein gel network as the KC concentration was below 0.75%. However, some relatively large lipid-rich microdomains were still observed in the protein gel networks, which are like the original state of non-digested oil droplets in the emulsion gel (Figure 2A). The low pK_a_ value (below 2.0) makes KC negatively charged and thus facilitates electrostatic attractions between negatively charged KC molecules and positively charged pea proteins at pH 3.0, thereby resulting in the formation of denser and compact gel network during gastric digestion [74,75]. Therefore, the addition of KC could effectively prevent the matrix breakdown of the protein gel network in the simulated stomach phase.

Structures of gastric digesta would possess a significant impact on the digestive behavior of emulsion gels during in vitro intestinal digestion [76]. Generally, all types of emulsion gels were further broken into smaller gel particles, due to the enzymatic hydrolysis of trypsin and lipase. As KC concentration was below 0.25%, pea protein emulsion gels were broken down in the initial stage of intestinal digestion, resulting in the release of individual or coalescence of oil droplets. As the concentration of KC was above 0.5%, gel particle size decreased gradually with digestion time, but gel fractions retained some structure even after 2 h. It should be noted that coalescence was observed between the adjacent oil droplets inside the gel fragments as the KC concentration was above 0.75%. This is possibly attributed to the immobilization of oil droplets by the gel networks, and some coalescence of oil droplets occurred between neighboring oil droplets caused by the displacement of interfacial proteins or peptides by bile salts facilitating the coalescence of oil droplets [73]. These results indicated that KC could modulate the microstructure of digesta within the simulated intestinal digestion.

#### 3.7.2. Size of Oil Droplet in the Digestive Fluids

The mean size of oil droplets within the digestive fluids is shown in Figure 8. The mean droplet size of the pea protein emulsion gels was around 34 μm after 1 h gastric digestion, indicating the occurrence of oil droplet coalescence due to the enzymatic hydrolysis of interfacial protein of oil droplets by pepsins [77]. The addition of KC at 0.25~0.75% resulted in a decrease in oil droplet size, with values of 16~21 μm. However, the large oil droplets with 36 and 108 μm were observed at the KC concentration of 1.0% and 1.5%, respectively. The diverse effects of KC on the droplet size in the gastric fluids may be attributed to several physicochemical mechanisms. At the low KC concentration, negatively charged KC molecules could adsorb to the surface of pea protein-coated oil droplets through the electrostatic attraction, thereby preventing the aggregation and coalescence by increasing the steric and electrostatic repulsion between oil droplets [39]. At the high KC concentration, the presence of excessive unabsorbed KC molecules in the digestive fluids caused extensive aggregation, flocculation, and coalescence due to a bridging or depletion mechanism even in the presence of a high droplet charge [78,79]. With further digestion in the intestinal stage, the mean size of oil droplets in the digestive fluids further gradually decreased due to the lipolysis and kept their size varied from 12~22 μm, which is consistent with previous observations of O/W water emulsions formulated with protein and anionic polysaccharides under gastrointestinal conditions [80].

#### 3.7.3. Protein Hydrolysis

The degree of protein hydrolysis in all formulations gradually increased with the digestion time during the continuous simulated gastrointestinal tract (Figure 9). The addition of KC could delay protein hydrolysis, which was more pronounced at the higher KC concentration in the intestinal stage. This effect could be attributed to the protective action of the KC network matrix (Figure 7), which created a barrier against penetration by the pepsin and trypsin [81]. Particularly, protein networks remained at 0.75~1.5% KC during the simulated intestinal phase (Figure 7), thereby preventing the erosion, degradation, and release of protein to the digestive fluids. Generally, the protein hydrolysis results were consistent with CLSM results (Figure 7).

#### 3.7.4. Free Fatty Acids Release

During the simulated intestinal digestion process, free fatty acids are produced by lipid hydrolysis, which exists in a mixed micelle form stabilized by bile salts in digestive fluids [82]. Figure 10 shows the release of free fatty acids from pea protein emulsion gels at various concentrations of KC during the in vitro intestinal digestion stage. Overall, a rapid increase in free fatty acids was observed during the initial stage, followed by a more gradual increase at longer times. In the absence of KC, around 27% of free fatty acids were released within the first 20 min. After that, the free fatty acid release rate slowed down with a value of 46% at the end of the intestinal digestion. The addition of 0.25% KC could improve the free fatty acid release rate and extent during the whole digestion. The increased release rate can be attributed to the fast breaking of the gel matrix and smaller oil droplet size (Figure 7 and Figure 8), facilitating lipase molecules’ absorption onto the surface of oil droplets for lipid hydrolysis [80]. With the further increasing concentration of KC, the lipid digestion of emulsion gels gradually slowed down. The significant prevention of lipid digestion was obverted at the 1.5% KC, with values of 13% and 37% after digestion for 20 min and 240 min, respectively. The inhibition of lipid hydrolysis at the high KC concentration could be associated with the retention of large gel particles and oil droplets retention (Figure 7 and Figure 8). The hard and dense gel networks can inhibit digestive enzymes and bile salts from accessing the gel interior, and extensive flocculation and coalescence could restrict the access of lipase molecules to oil droplet surfaces [73]. Moreover, the high KC concentration would increase the viscosity of gastrointestinal fluids and contribute to the less accessibility of lipase to oil droplets [83]. Overall, the release profile of free fatty acid is associated with changes in the microstructure and oil droplet size during the digestive process.

## 4. Conclusions

In this study, pea protein/κ-carrageenan composite emulsion gels have been successfully fabricated with a Ca^2+^-induced gelation method, and the relationship among the microstructure, texture, and digestive properties at various κ-carrageenan concentrations has been depicted (Figure 11). The microstructure characteristics of pea protein/κ-carrageenan composite emulsion gels are highly dependent on the κ-carrageenan concentration, which was mainly driven by hydrophobic interactions and hydrogen bonds. The addition of κ-carrageenan ranging from 0.25% to 0.75% facilitated the formation of compact and homogenous protein/polysaccharide coupled gel networks, increasing water holding capacity and lipid hydrolysis rate and extent. κ-Carrageenan at above 1.0% resulted in denser polysaccharide networks surrounding proteins and oil droplets, thereby increasing the gel harness and slowing down the gel degradation, free fatty acid release, and protein hydrolysis. These results indicated that it was possible to regulate the microstructure, texture property, and digestive behavior of the pea protein emulsion gels by the combination use of calcium and κ-carrageenan during the manufacturing process, which may be advantageous for the fat replacement, texture modification, and controlled release of nutrients in producing reformulated and plant-based alternative meat products.

## Figures and Tables

**Figure 1 foods-13-00483-f001:**
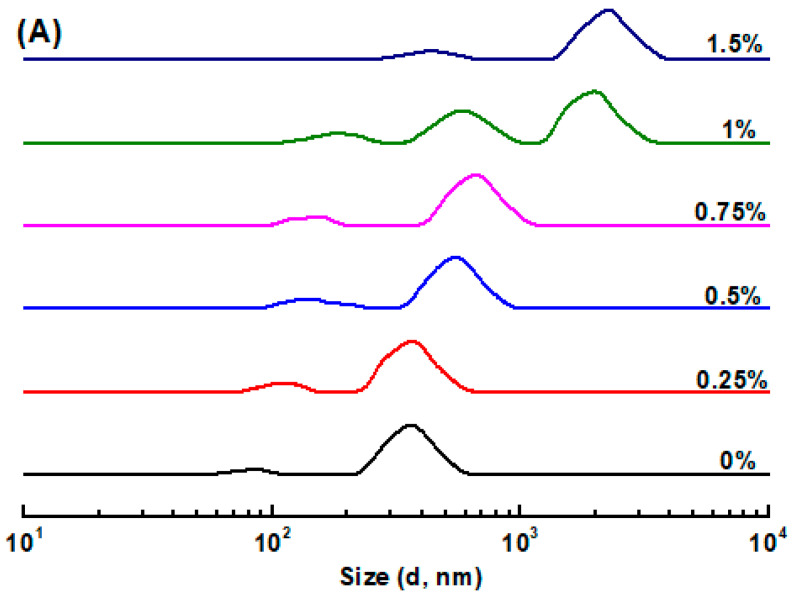
The size distribution (**A**) and ζ-potential (**B**) of pea protein isolate aggregate stabilized emulsions in the absence and presence of κ-carrageenan at various concentrations. Different letters (a–d) represent the statistically significant difference (*p* < 0.05).

**Figure 2 foods-13-00483-f002:**
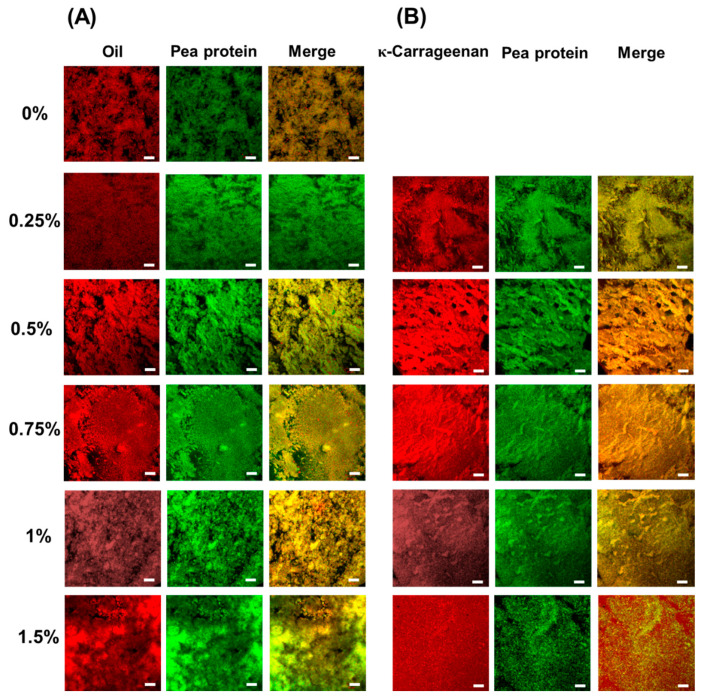
The confocal laser scanning microscopy images of pea protein emulsion-filled gels at various concentrations of κ-carrageenan. (**A**) The red color represents oil and the green color represents protein. (**B**) The red color represents κ-carrageenan and the green color represents protein. The scale bar is 20 μm.

**Figure 3 foods-13-00483-f003:**
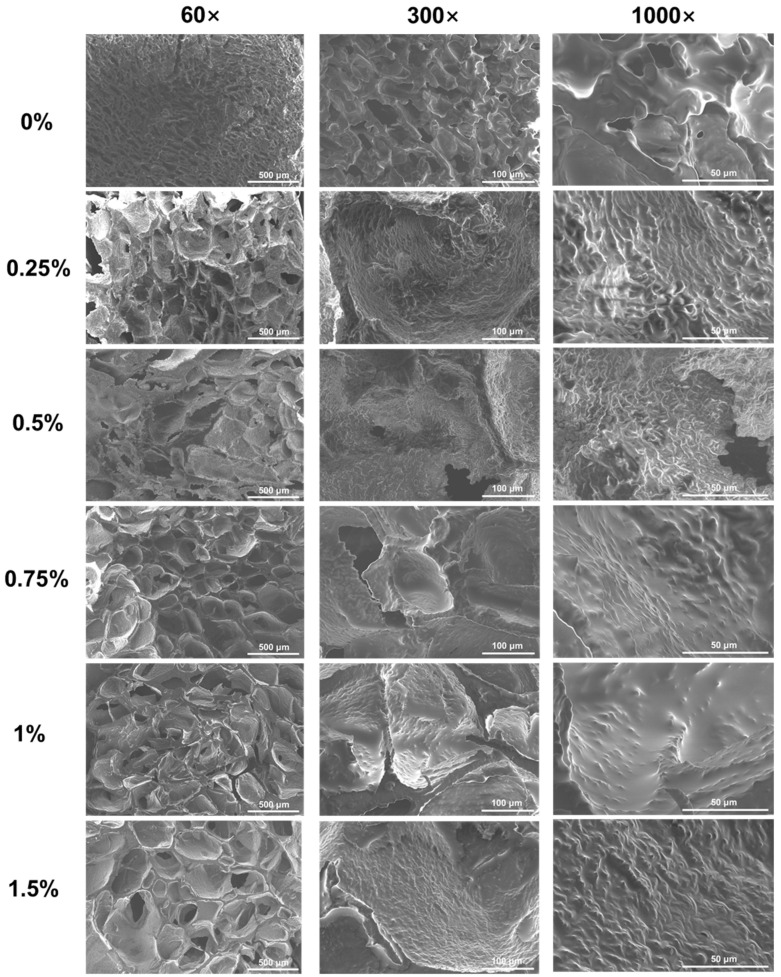
The scanning electron microscope images of freeze-dried pea protein emulsion-filled gels at various concentrations of κ-carrageenan.

**Figure 4 foods-13-00483-f004:**
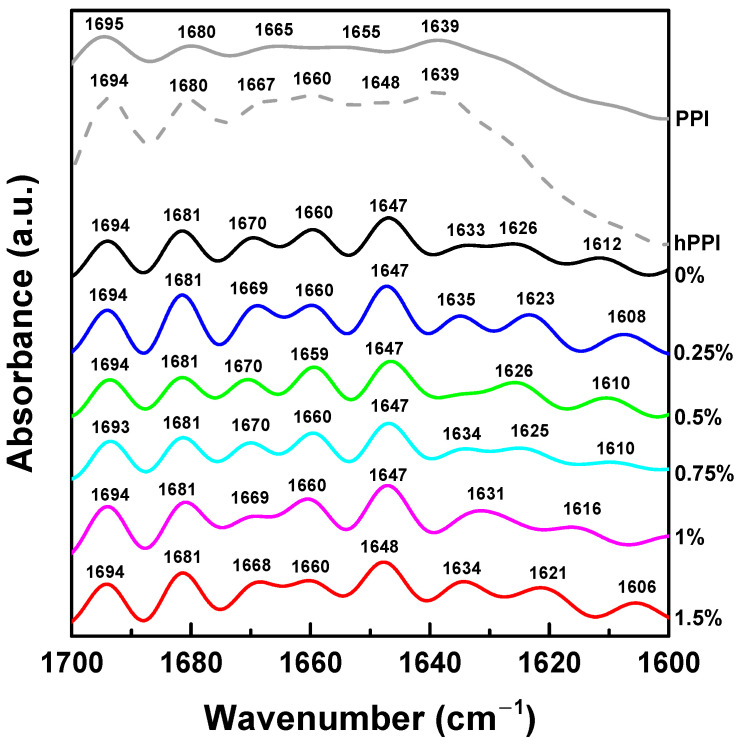
Deconvoluted Fourier-transform infrared (FT-IR) spectra in the amide I region of pea protein isolate (PPI), thermally denatured pea protein isolate (hPPI), and pea protein emulsion-filled gels at various concentrations of κ-carrageenan.

**Figure 5 foods-13-00483-f005:**
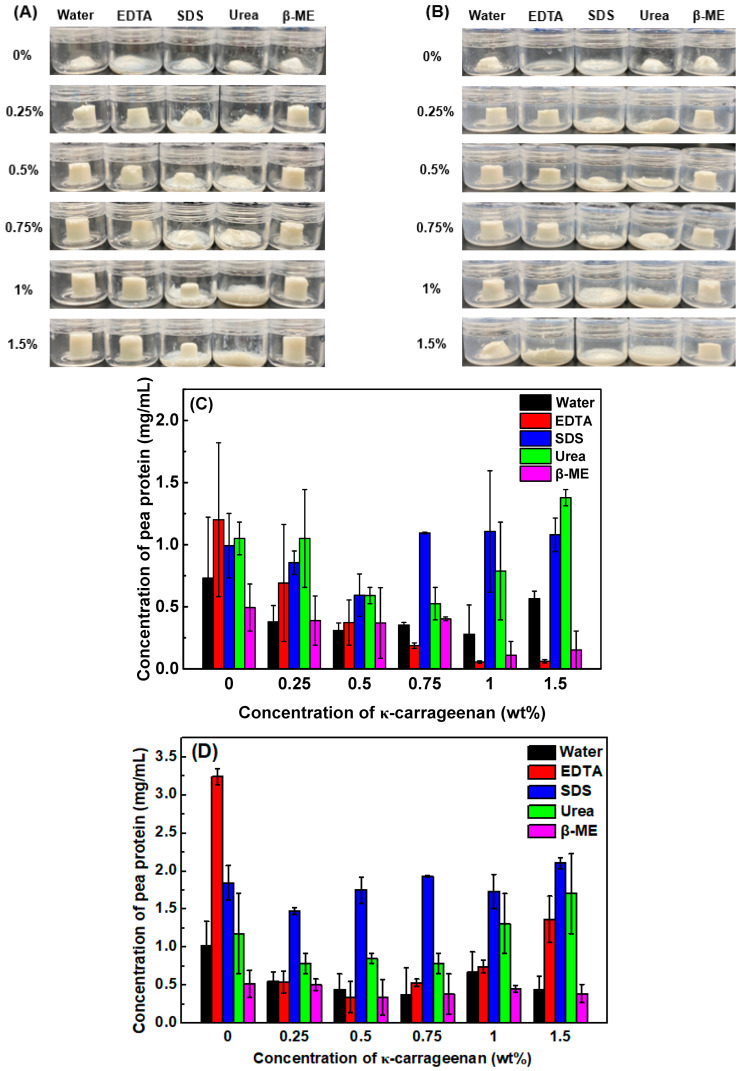
The visual appearance (**A**,**B**) and pea protein solubility (**C**,**D**) of the emulsion gel at various concentrations of κ-carrageenan in water, Na_2_•ethylene diamine tetraacetic acid (EDTA), sodium dodecyl sulfate (SDS), urea, β-mercaptoethanol (β-ME) at 25 °C (**A**,**C**) and 60 °C (**B**,**D**).

**Figure 6 foods-13-00483-f006:**
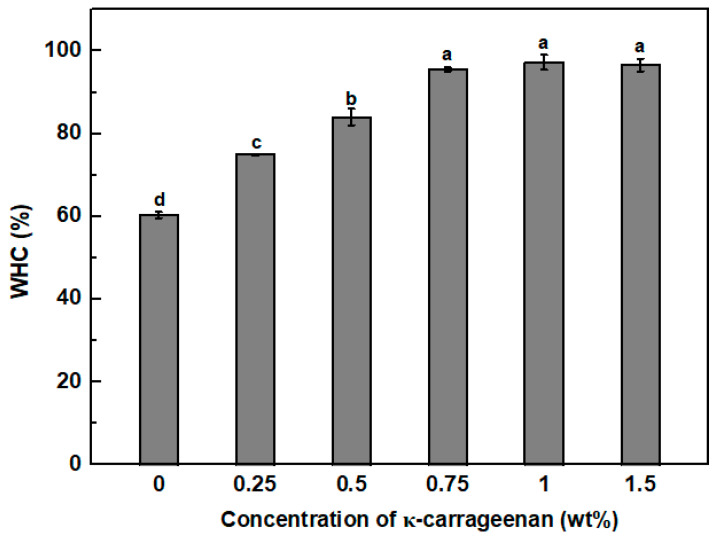
Water holding capacity (WHC, %) of pea protein emulsion-filled gels with various concentrations of κ-carrageenan. Different letters (a–d) represent the statistically significant difference (*p* < 0.05).

**Figure 7 foods-13-00483-f007:**
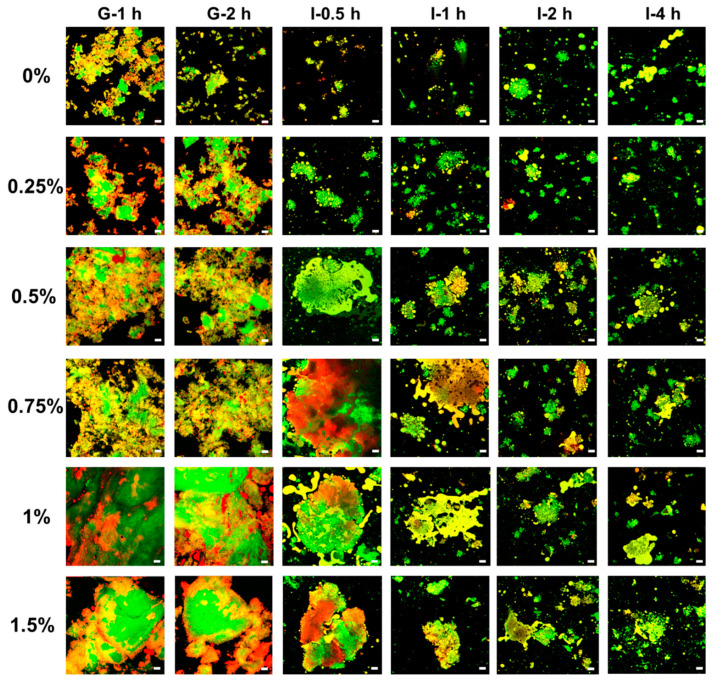
The microstructure of pea protein emulsion-filled gels after in vitro gastric (G) and intestinal (I) digestion with various concentrations of κ-carrageenan. The red color represents oil and the green color represents protein. The scale bar is 60 μm.

**Figure 8 foods-13-00483-f008:**
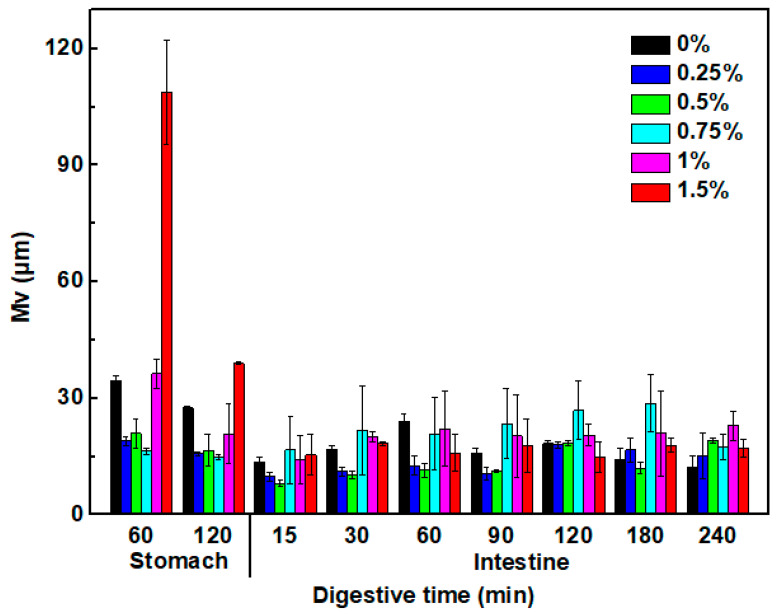
The mean size (Mv, μm) of pea protein emulsion-filled gels digested in vitro with various concentrations of κ-carrageenan.

**Figure 9 foods-13-00483-f009:**
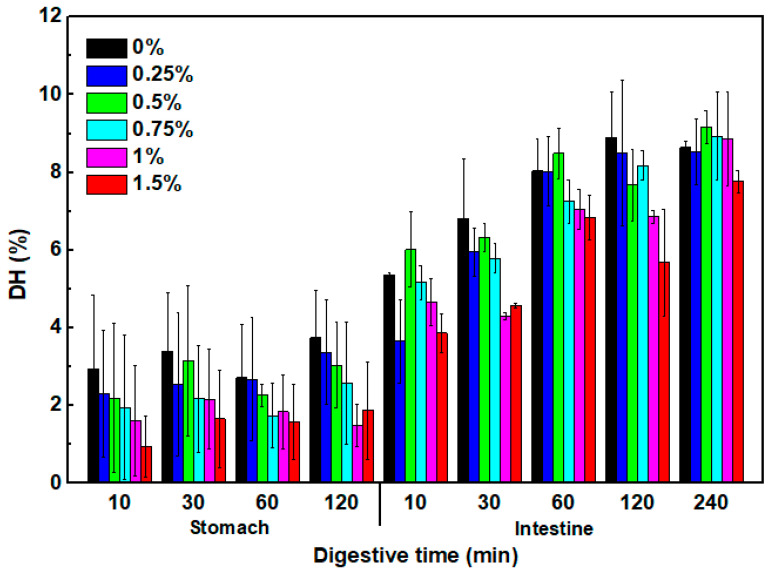
The protein hydrolysis degree (DH, %) of pea protein emulsion-filled gels digested in vitro at various concentrations of κ-carrageenan.

**Figure 10 foods-13-00483-f010:**
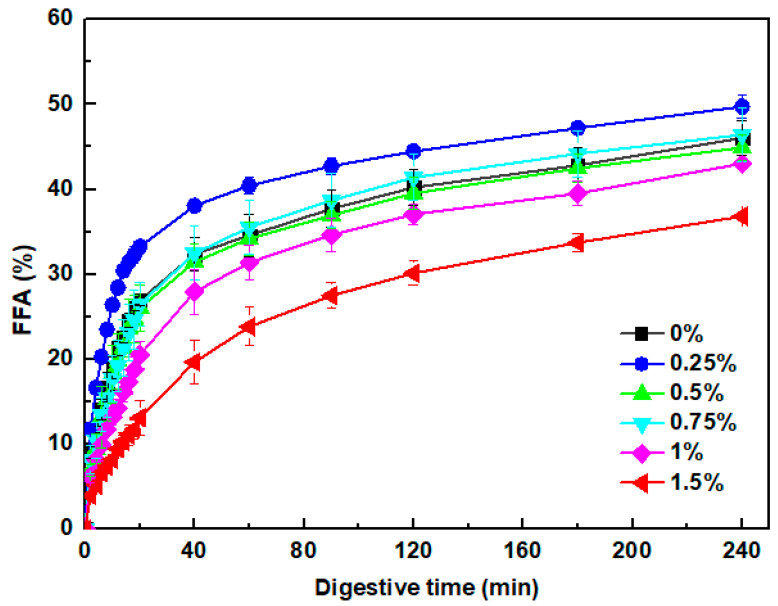
The release of free fatty acid (FFA, %) of pea protein emulsion-filled gels digested in vitro at various concentrations of κ-carrageenan.

**Figure 11 foods-13-00483-f011:**
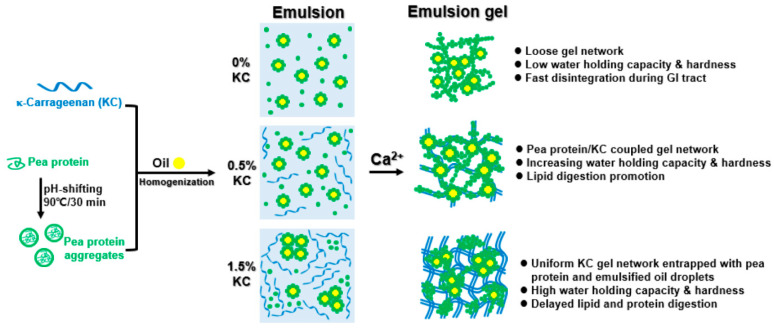
An illustration of formation mechanism and possible structure–property relationship of pea protein emulsion-filled gels at various concentrations of κ-carrageenan.

**Table 1 foods-13-00483-t001:** The texture profile analysis data of pea protein emulsion-filled gels at various concentrations of κ-carrageenan.

κ-Carrageenan Concentration (%)	Hardness (g)	Springiness	Cohesiveness	Chewiness (g)
0	20.956 ± 4.406 ^d^	0.366 ± 0.044 ^c^	0.307 ± 0.038 ^bc^	3.140 ± 0.440 ^c^
0.25	43.804 ± 3.404 ^cd^	0.531 ± 0.007 ^ab^	0.421 ± 0.003 ^a^	7.060 ± 3.710 ^c^
0.5	214.437 ± 88.419 ^c^	0.530 ± 0.078 ^ab^	0.370 ± 0.016 ^ab^	25.010 ± 2.150 ^c^
0.75	215.681 ± 13.635 ^c^	0.560 ± 0.001 ^a^	0.261 ± 0.021 ^c^	31.199 ± 4.029 ^c^
1	767.646 ± 211.026 ^b^	0.442 ± 0.077 ^bc^	0.396 ± 0.100 ^ab^	125.333 ± 24.603 ^b^
1.5	1560.162 ± 51.637 ^a^	0.506 ± 0.026 ^ab^	0.270 ± 0.021 ^c^	215.448 ± 35.271 ^a^

Different letters in the same column indicate statistically significant differences (*p* < 0.05).

## Data Availability

Data is contained within the article.

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
