# Peer review of "Impact of κ-Carrageenan on the Cold-Set Pea Protein Isolate Emulsion-Filled Gels: Mechanical Property, Microstructure, and In Vitro Digestive Behavior"

_foods, 2024, doi:10.3390/foods13030483_

Round 1
Reviewer 1 Report
Comments and Suggestions for Authors
To whom it may concern,
The current manuscript entitled "Impact of κ-carrageenan on the cold-set pea protein isolate emulsion-filled gels: Mechanical property, microstructure, and in vitro digestive behavior" is a well-written manuscript and deserves for further consideration. Some minor improvements are needed as shown in the pdf. The plaragism by Turnitin is also 30% which is recommended to be less than 20%.
Thank you

Comments on the Quality of English LanguageAuthor Response
We appreciate the comments and corrections from the editor and reviewers and would like to thank him/her for taking the time to study our paper. In the following, we will reply to his/her comments point-by-point.
Comments from the reviewers:
-Reviewer 1
To whom it may concern,
The current manuscript entitled "Impact of κ-carrageenan on the cold-set pea protein isolate emulsion-filled gels: Mechanical property, microstructure, and in vitro digestive behavior" is a well-written manuscript and deserves for further consideration. Some minor improvements are needed as shown in the pdf. The plaragism by Turnitin is also 30% which is recommended to be less than 20%.
Thank you
Authors' response: Thank the reviewer for invaluable suggestions. We have revised and complemented the manuscript according to the reviewer's suggestions. The specific response to comments was answered point by point.
Although the overall similarity index obtained by Turnitin was around 30%, the similarity index for each specific reference did not exceed 1%. Furthermore, the most similarity was observed in the section on materials and methods. A few similarities could be observed in the section's introduction and results and discussion. We have reorganized these sentences for the decrease in similarity index.
Line 19 Suggest to replace by "More understanding on the ......."
Authors' response: Thank you for your valuable comments. The sentence "Clarification of the relationship among the microstructure ……" has been changed to "More understanding on the relationship among the microstructure ……" in line 19 of the revised manuscript.
Line 19 The methodology is unclear. Please provide more explanations on the treatments.
Authors' response: Thank you for your valuable comments. More explanations on the treatment and analysis methods have been added in the abstract of the revised manuscript.
Line 93 Please provide a statement of problem before the objectives. What is the current gap?
Authors' response: Thank the reviewer for invaluable suggestions. Although pea protein isolate has been used for the fabrication of emulsion gels, there has not been a systematic investigation on the structure-property relationships of pea protein-based emulsion gels affected by the addition of κ-carrageenan. A statement of the problem has been provided before the objectives in lines 94-96 of the revised manuscript.
Line 113 The unit should be "V" which stands for volume. Because water is liquid
Authors' response: Thank the reviewer for the correction. The text "1:40 (w/w)" has been changed to "1:40 (w/v)" in line 116 of the revised manuscript.
Line 165 Please mention the magnification.
Authors' response: Thank you for your valuable comments. The magnifications we have used are 60 ×, 300 × and 1000 ×. It has been supplemented in line 174 of the revised manuscript.
Line 193 How much mL?
Authors' response: Thanks for the reviewer's careful review. The volume of SGF we used was 20 mL, which has been added in line 198 of the revised manuscript.
Line 245 Please elaborate more. These results should be discussed.
Authors' response: Thank you for your valuable suggestions. Further explanation and discussion have been supplemented in lines 250-255 of the revised manuscript.
Line 272 when
Authors' response: Thank the reviewer for the correction. The text "as" has been changed to "when" in line 282 of the revised manuscript.
Line 307 I noticed that you used 3 magnifications: 60, 300 and 1000. Please provide some reasons why you didn't use i magnification, for example 1000
Authors' response: Thank the reviewer for the question. The selection of multiple magnifications will be beneficial to the observation of gel structures at different scales. The morphology of the overall structure of the emulsion gels can be observed at a magnification of 60 ×, while a larger magnification can be used to observe and compare more detailed morphological features, such as the pore size, the inner wall of the pore, and so on.
Line 396-397 What could be the reason?
Authors' response: Thank the reviewer for the question. On the one hand, emulsion gels possessed high water holding capacity when the κ-carrageenan concentration was 0.75%. On the other hand, the further increase in the κ-carrageenan did not damage the gel structure. Therefore, the further increase in the κ-carrageenan concentration exhibited no significant impact on the water-holding capacity of emulsion gels. The text “attributing to the fact that the high KC concentration did not damage the gel structure (Fig. 3)” in lines 408-409 of the revised manuscript.
Line 545 Please mention that what concentration showed the best result.
Authors' response: Thank you for your suggestions. In this study, we aim to investigate the effect of κ-carrageenan on the properties of pea protein emulsion gels and the relationship among the structure, texture, and digestive properties. The addition of κ-carrageenan could significantly improve the gel strength and water-holding capacity. However, the microstructure and digestive properties were highly dependent on the polysaccharide concentration. For example, the composite emulsion gels exhibited a soft texture and fast hydrolysis of protein and lipids as the κ-carrageenan concentration below 0.5%, while κ-carrageenan concentrations above 1% could effectively slow down the digestion of lipids. These tunable properties can meet the requirements of different food systems. Therefore, it is hard to say which concentration is the best result. We have reorganized conclusions and supplemented some specific applications of the pea protein/κ-carrageenan emulsion gel according to Reviewer 2’s suggestions.

Reviewer 2 Report
Comments and Suggestions for Authors
Clearly written report on a rather thorough study of a specific, innovative gel suitable for food industry. Motivation, procedures are clear, all results presented and adequately discussed. Below I give some comments which can still improve readability but otherwise I can endorse accepting tjis text for publication in this journal.
1. L. 122/123 – what were the titrants?
2. L. 139 – at least one sentence informing on the basis of the gelation method would be very useful for readers.
3. Capture to Fig. 3 should stress that the images are of freeze-dried, not native gels.
4. The vertical axis in Figure 5C should be better cut at 2 mg/ml to improve readability; the difference to 5D will remain evident.
5. Table 1 – I do not think that all decimal places, especially for the standard deviations, are significant.
6. Part 3.7.4 – readers could wonder if the released fatty acids are supposed to be water soluble (soluble in the test medium) or still emulsified.
7. Perhaps some specific applications could be envisioned in the end of Conclusions.
8. Carrageenan is a basic constituent of the studied gels – are there no more characteristics on the preparation used in this work (provided by the supplier)?
Author Response
We appreciate the comments and corrections from the editor and reviewers and would like to thank him/her for taking the time to study our paper. In the following, we will reply
-Reviewer 2
Clearly written report on a rather thorough study of a specific, innovative gel suitable for food industry. Motivation, procedures are clear, all results presented and adequately discussed. Below I give some comments which can still improve readability but otherwise I can endorse accepting tjis text for publication in this journal.
Authors' response: Thank the reviewer for invaluable suggestions. We have revised and complemented the manuscript according to the reviewer's suggestions. The specific response to comments was answered point by point.
- L. 122/123 – what were the titrants?
Authors' response: We used 6 M NaOH and 6 M HCl for pH adjustment of protein solutions, which has been added in lines 125-126 of the revised manuscript.
- L. 139 – at least one sentence informing on the basis of the gelation method would be very useful for readers.
Authors' response: Thank you for your valuable comments. We have complemented the gelation method in lines 142-144 of the revised manuscript. The sentence has been changed to “The emulsions in the absence and presence of KC were rapidly mixed with 1 M CaCl2 solution and stored overnight at 4°C for gel formation. The final concentration of CaCl2 was 40 mM in the gels”.
- Capture to Fig. 3 should stress that the images are of freeze-dried, not native gels.
Authors' response: Thank you for your valuable comments. The caption of Fig. 3 has been changed to “The scanning electron microscope images of freeze-dried pea protein emulsion-filled gels at various concentrations of κ-carrageenan” in lines 337-338 of the revised manuscript.
- The vertical axis in Figure 5C should be better cut at 2 mg/ml to improve readability; the difference to 5D will remain evident.
Authors' response: As requested, Fig. 5C has been revised in line 384 of the revised manuscript.
- Table 1 – I do not think that all decimal places, especially for the standard deviations, are significant.
Authors' response: Thank you for your valuable comments. Considering the accuracy of the texture data, we presented all decimal places from the texture analyzer. At the same time, we have kept the standard deviation at the same number of decimal places as the averages. Therefore, we would like to present all decimal places of texture data in the paper.
- Part 3.7.4 – readers could wonder if the released fatty acids are supposed to be water soluble (soluble in the test medium) or still emulsified.
Authors' response: Thank you for your suggestions. Free fatty acids produced by lipid hydrolysis are generally presented in a mixed micelle form stabilized by bile salts in digestive fluids (Food Chemistry, 2021, 348, 129148; Advances in Colloid and Interface Science, 2011, 165, 14-22). The text “During the simulated intestinal digestion process, free fatty acids are produced by the lipid hydrolysis, which exists in a mixed micelle form stabilized by bile salts in digestive fluids [82].” has been added in lines 520-522 of the revised manuscript.
- Perhaps some specific applications could be envisioned in the end of Conclusions.
Authors' response: Thank you for your valuable suggestions. As requested, the specific application of the pea protein/κ-carrageenan composite emulsion gel has been supplemented it in lines 560-562 of the revised manuscript.
- Carrageenan is a basic constituent of the studied gels – are there no more characteristics on the preparation used in this work (provided by the supplier)?
Authors' response: Thank you for your valuable questions. The κ-carrageenan used in this study was purchased from Sigma-Aldrich Co., Ltd. In this study, we focused on the effect of κ-carrageenan on the properties of pea protein emulsions. Therefore, no additional treatment and characterization on κ-carrageenan was performed. The basic information (e.g., solubility, color, and viscosity) could be observed on the website of the supplier. Furthermore, there have been many scientific studies on the physicochemical properties of κ-carrageenan from Sigma-Aldrich Co., Ltd (Biomacromolecules 2019, 20, 1731-1739; Journal of Texture Studies, 2019, 50(6), 520-538; Food Hydrocolloids, 2016, 61, 793-800). We have added the product number of κ-carrageenan in line 106 of the revised manuscript.
